# Allergic Diseases and Childhood Obesity: A Detrimental Link?

**DOI:** 10.3390/biomedicines11072061

**Published:** 2023-07-22

**Authors:** Camilla Stefani, Luca Pecoraro, Carl-Erik Flodmark, Marco Zaffanello, Giorgio Piacentini, Angelo Pietrobelli

**Affiliations:** 1Pediatric Unit, Department of Surgical Sciences, Dentistry, Gynecology and Pediatrics, University of Verona, 37126 Verona, Italymarco.zaffanello@univr.it (M.Z.); giorgio.piacentini@univr.it (G.P.); angelo.pietrobelli@univr.it (A.P.); 2Childhood Obesity Unit, Skåne University Hospital, 21550 Malmö, Sweden; carl-erik.flodmark@med.lu.se; 3Pennington Biomedical Research Center, Baton Rouge, LA 70808, USA

**Keywords:** obesity, childhood asthma, allergic rhinitis, allergic conjunctivitis, atopic dermatitis, food allergy, chronic urticaria

## Abstract

Several epidemiological studies have described childhood obesity as a risk factor for atopic disease, particularly asthma. At the same time, this association seems to be more conflicting for allergic rhinitis, atopic dermatitis, and chronic urticaria. This article aims to deepen the possibility of a relationship between childhood obesity and allergic diseases. As regards asthma, the mechanical and inflammatory effects of obesity can lead to its development. In addition, excess adiposity is associated with increased production of inflammatory cytokines and adipokines, leading to low-grade systemic inflammation and an increased risk of asthma exacerbations. Allergic rhinitis, atopic dermatitis, food allergies, and chronic urticaria also seem to be related to this state of chronic low-grade systemic inflammation typical of obese children. Vitamin D deficiency appears to play a role in allergic rhinitis, while dyslipidemia and skin barrier defects could explain the link between obesity and atopic dermatitis. Starting from this evidence, it becomes of fundamental importance to act on body weight control to achieve general and allergic health, disentangling the detrimental link between obesity allergic diseases and childhood obesity. Further studies on the association between adiposity and atopy are needed, confirming the biologically active role of fat tissue in the development of allergic diseases and exploring the possibility of new therapeutic strategies.

## 1. Introduction

Obesity and allergic diseases are both significant public health issues. Obesity in children is defined by a body mass index (BMI) above the 95th percentile for age and sex, while overweight is defined by a BMI between the 85th and 95th percentile for age and sex [1]. In Europe, the prevalence of obesity is rising in almost all countries; in some of these, it has nearly doubled in the last four decades [2,3]. At the same time, the prevalence of allergic diseases, such as allergic rhinitis (AR), atopic dermatitis (AD), bronchial asthma (BA), food allergies (FA), and chronic urticaria (CU), is also increasing considerably. AR is the most common allergic disease, affecting 10 to 30 percent of children and adults in the United States and other industrialized countries [4]. AD affects up to 12% of children and 7.2% of adults, leading to high healthcare use [5]. BA is one of the most common chronic diseases in childhood, with a prevalence of 6% to 9% and an incidence that is continuously increasing worldwide [6]. Regarding FA, the prevalence is 6.7% in the USA [7]. Up to 1% of the population in the USA and Europe suffer from chronic urticaria during their lifetime. Both children and adults can develop CU, but it is more common in adults [8]. There is rising epidemiological evidence that obesity increases the risk of allergic diseases, particularly asthma [9,10]. Studies in monozygotic and heterozygotic twins suggest that eight percent of the genetic component of obesity is shared with asthma [11]. In addition, it is known from several studies that children of mothers with atopy are at greater risk of developing atopy themselves [12]. Allergic diseases result from the interaction between genetic and environmental factors, which may aggravate the susceptibility to allergic diseases through epigenetic modification. Specifically, obesity represents an environmental factor involved in immunological changes resulting in switching the immune system toward a Th2-cytokine profile and a high risk of allergy [12]. In addition, intrauterine exposure to tobacco smoke seems to play a role in developing obesity and asthma in adult life. Penn and colleagues demonstrated that tobacco smoking exposure in mice during fetal life exacerbates subsequent adult responses to initial allergen exposure by increasing bronchial hyperactivity [13]. The studies on the association between obesity and atopy, however, have conflicting results. Some data indicate the correlation of a higher BMI with an increased prevalence of atopy [14,15], while others show a lack of relation between the serum IgE level and blood eosinophil percentage with obesity [16,17]. Some recent evidence has also suggested the role of obesity in developing chronic rhinosinusitis (CRS) with nasal polyposis [18,19]. However, the effect of obesity on sino-nasal inflammation remains controversial [20,21]. This article aims to deepen the possibility of a relationship between childhood obesity and allergic diseases.

## 2. Obesity and Asthma

Epidemiological studies have demonstrated the connection between asthma and obesity. Although the relationship between asthma and obesity remains not fully explained from a pathophysiological point of view, it has been demonstrated that obesity is a risk factor for asthma [9,10,11,12,13,14,15,16,17,18,19,20,21,22,23]. Rönmark and colleagues found that obesity increased the risk of asthma by 2.7-fold and the risk of being overweight by 2-fold compared with patients of normal weight [24]. A meta-analysis including six prospective studies found that obese children have a two-fold higher risk for asthma than normal-weight children [25]. The link between asthma and obesity can be explained partly by mechanical factors and partly by the chronic low-grade inflammation accompanying obesity. In some patients, obesity precedes asthma; in others, asthma precedes obesity, suggesting that asthma or asthma treatment may also be a risk factor for obesity [26,27]. Asthma occurs more frequently in obese patients, who tend to have more symptoms, increased asthma severity, healthcare use, and worse quality of life [28,29]. Different studies suggest that asthma in obese patients may differ from the classical phenotype of the disease, showing a non-Th2-mediated response. Asthma exacerbations in obese subjects frequently present a reduced response to standard medications [16,17,18,19,20,21,22,23,24,25,26,27,28,29,30]. This severe asthma phenotype characterized patients, predominantly females, without eosinophilic airway inflammation [31]. Lessard and colleagues found a significant inverse correlation between the percentage of eosinophils in induced sputum and BMI and waist circumference [16]. This may suggest a role of abdominal fat in the noneosinophilic inflammation of the airway, typical of obese subjects. Several studies have focused on the relationship between BMI and exhaled nitric oxide (eNO), a measure of airway inflammation, with conflicting results. Some authors describe a positive association between eNo concentration and BMI, concluding that eNO can be a systemic link between airway inflammation and obesity [32]. Other studies show a negative correlation between BMI and eNO in obese patients with asthma [33]. Several factors related to obesity likely contribute to the increased risk of asthma in children, but the exact pathogenetic mechanism is still unknown. A lot of factors seem to be involved, primarily, dietary factors. Diets rich in sugar and saturated fatty acids and low in fiber and antioxidants increase the risk of obesity and respiratory symptoms [34]. Low vitamin D levels may also contribute to the risk of asthma in obese patients [35]. Dyslipidemia and insulin resistance are associated with impaired forced vital capacity (FVC) and asthma severity. Insulin is a trophic stimulus for low airway smooth muscle cells; it induces laminin production, leading to muscle hypertrophy. It also enhances airway hyperreactivity via stimulating parasympathetic innervation, thus promoting airway obstruction [36]. High levels of total cholesterol (TC) and low-density lipoprotein (LDL) are more common in obese asthmatic children and are associated with lower lung function [37]. The exact mechanism by which dyslipidemia acts on pulmonary function is not yet known. Maternal obesity before and during pregnancy also appears to play a role in children’s subsequent asthma development. McDonald and colleagues deepened a murine model to investigate the effect of maternal diet on adult offspring bronchial hyperactivity. They found that a diet rich in saturated fatty acids during pregnancy plays a key role in developing airway hyperactivity [38]. The higher risk of allergic reactions in children of obese mothers seems to depend on the increased production of inflammatory cytokines induced by excess adipose tissue. Such changes are likely to result from long-lasting changes in miR-155 and miR-133b expression [39]. Secondly, environmental factors: different studies suggest that exposure to air pollution and passive tobacco smoking are independent risk factors for the development of both asthma [40,41] and obesity in children [42,43]. Thirdly, impaired lung growth: children with obesity have an incongruence between the development of the lungs and the airway. They have increased lung volume relative to airway caliber (“dysanapsis”), reflected by a lower ratio of forced expiratory volume in one second to forced vital capacity (FEV1/FVC), despite normal values for FEV1 and FVC. Dysanapsis is associated with airflow reduction, increased asthma exacerbations, and the use of systemic glucocorticoids in obese children [44]. Fourthly, mechanical factors: obesity causes substantial changes to the mechanics of the lungs and chest wall, and these mechanical changes cause asthma and asthma-like symptoms such as dyspnea, wheezing, and airway hyperresponsiveness. Excess fat mass in the chest wall and abdomen reduces the functional residual capacity of the lung (FRC) [37]. It is also associated with both reduced forced vital capacity (FVC) and forced expiratory volume in one second (FEV1) [37,38,39,40,41,42,43,44,45]. Low tidal volume breaths due to thoracic and abdominal fat excess lead to low lung volumes [46], causing alveolar hypoventilation and an increase in airway resistance [47]. These events, in turn, cause airway hyperresponsiveness, resulting in higher respiratory rates and increased work when breathing. The respiratory system compliance is also reduced, mainly due to the increased elastic resistance of the chest wall [48]. Together, these modifications cause a stiffening of the airway smooth muscle in obese subjects, leading to narrowed airways and a reduced broncho-dilatory effect [49]. Fifthly, the immune cell function: traditionally, asthma and other atopic diseases are associated with T-helper 2 cellular inflammation with elevated levels of IgE and eosinophils. Adaptive and innate immune cell functions are altered in obesity. Several studies show the suppression of Th2 lymphocyte function in obese patients [50,51]. Obesity polarizes the immune response toward Th1 rather than the classical Th2 [52,53]. Other studies show a polarization toward Th17 immune response in obese patients. Visceral inflammation with increased proinflammatory macrophages (M1) also occurs in obese asthmatics and may determine systemic inflammation and asthma severity [54]. In obese subjects, oxidative stress, products of cell necrosis, and the overload of free fatty acids lead to polarization toward an M1 phenotype, while the anti-inflammatory M2 macrophages are reduced [55]. M1 macrophages induce airway obstruction in obese patients [56]. Eosinophil function is also altered by obesity. While submucosal eosinophils are increased in obese patients with asthma compared to lean subjects, eosinophils in peripheral blood and sputum are not raised with obesity [57]. All these aspects could explain why current asthma medications, including corticosteroids, leukotriene inhibitors, and biological agents against Th2 response and eosinophils, are less effective for obese asthmatic patients [37]. Lastly, the adipose tissue mediators: adipose tissue is recognized as an active endocrine organ that can affect the function of other organs and as an important source of proinflammatory cytokines, chemokine, and growth factors [58,59]. Excess adiposity is associated with increased production of inflammatory cytokines (IL-6, IL-1ß, and TNF-alpha), leading to low-grade systemic inflammation and increased risk of asthma exacerbations [12,13,14,15,16,17,18,19,20,21,22,23,24,25,26,27,28,29,30,31,32,33,34,35,36,37,38,39,40,41,42,43,44,45,46,47,48,49,50,51,52,53,54,55,56,57,58,59,60]. High levels of circulating IL-6 are associated with poor asthma control [61], and markers of inflammation, like C reactive protein (CRP) and fibrinogen, are increased in patients with asthma and obesity compared with those with asthma alone [12,13,14,15,16,17,18,19,20,21,22,23,24,25,26,27,28,29,30,31,32,33,34,35,36,37,38,39,40,41,42,43,44,45,46,47,48,49,50]. In recent years, several adipose tissue-derived cytokines have been characterized, the so-called adipokines (leptin, adiponectin, and resistin). Adipokines play a crucial role in energy homeostasis and inflammatory and immune response, most often promoting inflammation [55]. Leptin and resistin are proinflammatory, while adiponectin has mainly anti-inflammatory properties [62]. Leptin levels are positively correlated with adipose tissue mass, and leptin is therefore considered one of the factors explaining the connection between obesity and asthma [58]. The prominent role of leptin is the suppression of food intake via inhibiting hypothalamic nuclei that stimulate hunger and stimulation of those which induce satiety. Nevertheless, obese patients develop leptin resistance with a decreased sensitivity to anorexinergic stimuli. Leptin loss of function leads to hyperphagia, rapid weight gain, and insulin resistance [63]. Leptin also has immunomodulatory activities, stimulating neutrophil activation and chemotaxis, oxygen radical release, and the survival of macrophages, eosinophils, basophils, and natural killer cells [55]. IL6 and leptin downregulate the activity of regulatory T-lymphocytes (Tregs), decreasing immunological tolerance to antigens and thus increasing the risk of allergies and other immune-mediated diseases [12]. Adiponectin has anti-inflammatory properties and is associated with a lower risk of asthma, independent of BMI [59,60,61,62,63,64]. In macrophages, adiponectin promotes M2 phenotype polarization, TNF-alpha secretion reduction, and scavenger activity enhancement. Moreover, it stimulates the release of IL-10, one of the major anti-inflammatory cytokines, that plays a central role in regulating the immune response and improving insulin sensitivity [55]. Obesity is associated with lower adiponectin levels and an increased risk of cardiovascular diseases [65]. Resistin is mainly secreted by monocytes and macrophages, while a small fraction derives from adipocytes. The role of resistin in obesity and asthma remains unclear; there are only a few publications on this topic with conflicting results. Some studies found higher resistin levels in asthmatics, and its levels correlated with worse disease control [66], while other authors suggested a protective role of resistin against asthma [67]. In conclusion, adipokines are central to the link between obesity and altered immune response, leading to low-grade systemic inflammation and reduced immune tolerance. Weight loss after lifestyle intervention trials effectively reduces serum inflammatory markers and insulin resistance in obese children and adolescents, leading to better asthma control, lung function indices, and quality of life [37,38,39,40,41,42,43,44,45,46,47,48,49,50,51,52,53,54,55,56,57,58,59,60,61,62,63,64,65,66,67,68]. In conclusion, obesity may be associated with an asthma phenotype less responsive to classical anti-inflammatory treatment. In this context, a more useful intervention is to act on the reduction in body weight. Weight loss at any time in the individual’s lifespan has been associated with improved asthma symptoms and quality of life [37].

## 3. Obesity and Allergic Rhino-Conjunctivitis

Several studies have investigated the correlation between obesity and AR and rhino-conjunctivitis. A recent meta-analysis of 30 observational studies found a statistically significant association between obesity and the risk of AR in children [69]. The authors hypothesized that this link may derive from a common underlying inflammatory etiology. Immunological alterations caused by obesity can lead to reduced immunological tolerance to antigens, thus increasing the risk of AR. However, the association between obesity and AR may also be determined by other factors, such as vitamin D deficiency: obese patients with low vitamin D levels seem to have an increased risk for atopic disorders [70]. Luo and colleagues found a positive association between obesity and atopic dermatitis and rhinitis in adults: the associations of obesity with atopic dermatitis (OR = 2.7, 95% CI: 1.2, 6.3) and atopic rhinitis (OR = 3.1, 95% CI: 1.1, 8.7) were both statistically significant [71]. Other studies have shown a positive correlation between the increase in BMI and the prevalence of allergic rhinitis and conjunctivitis, especially among young female adults and children [15,16,17,18,19,20,21,22,23,24,25,26,27,28,29,30,31,32,33,34,35,36,37,38,39,40,41,42,43,44,45,46,47,48,49,50,51,52,53,54,55,56,57,58,59,60,61,62,63,64,65,66,67,68,69,70,71,72]. Han and colleagues investigated the mechanisms by which obesity affects the severity of AR. They measured leptin and inflammatory biomarker levels in the serum to analyze the correlation with the severity of AR. They found that IL-1β, a biomarker of active inflammation, was significantly higher in individuals with AR than in those without and in obese subjects compared to the normal weight group. Data analysis revealed leptin levels were associated with increased IL-1β expression in children with AR. In the multivariate analysis, elevated leptin (11.3-fold increase in risk) and high expression of IL-1β (5.8-fold increase in risk) emerged as significant risk factors for moderate to severe persistent allergic rhinitis [73]. Other recent studies suggest the role of leptin in the development of AR in obese children. Leptin seems to upregulate the expression of type 2 innate lymphoid cells (ILC2s), a newly discovered population of innate immune cells that may be involved in AR development [74,75]. These results suggest that obesity is a considerable risk factor for the exacerbation and severity of symptoms of AR. On the other hand, Kusunoki and colleagues found a negative association between higher BMI and allergic conjunctivitis and allergic rhinitis (*p* < 0.0001), especially among boys [76]. Overall, this study showed a positive association between obesity and atopic dermatitis and a negative association between obesity and allergic rhino-conjunctivitis, especially in boys. Han and colleagues performed a cross-sectional study of obesity indicators and rhinitis using data from 8165 participants in the 2005–2006 National Health and Nutrition Examination Survey. Multivariate regression assessed the relationship between obesity and rhinitis in children and adults. They found that obesity was associated with an increased risk of nonallergic rhinitis but not AR in adults, while in children, it was associated with a reduced risk of AR, regardless of sex [77]. As allergic rhinitis and conjunctivitis are presumed to be pure Th2-type diseases, it can be argued that obesity works by inhibiting the Th2 response in these pathologies, reducing their prevalence. In contrast, atopic dermatitis, which has a more complex etiology involving the Th1 response, can be exacerbated by obesity. Allergic rhinitis is often the prerequisite for the subsequent development of asthma (the so-called “atopic march”). In this regard, it becomes of fundamental importance to act on body weight control to prevent allergic rhinitis and the subsequent evolution into bronchial asthma [78].

## 4. Obesity and Nasal Polyps

In recent years, several studies have investigated the association between obesity and the development of chronic rhinosinusitis (CRS) with nasal polyps. A recent prospective population-based study conducted in Norway using questionnaire-based data (*n* = 5769) analyzed the correlation between BMI and the risk of new-onset CRS between 2013 and 2018. The researchers found that the odds ratio of new-onset CRS was 53% higher [OR 1.53 (1.11–2.10)] in the obese group compared to the normal-weight group [18]. BMI, therefore, seems to be an important risk factor for the development of CRS and should always be evaluated when assessing a patient for CRS. Other cross-sectional studies have demonstrated the link between obesity and CRS. Nam and colleagues analyzed a population of 32,384 subjects aged between 19 and 86 years, showing that the prevalence of CRS with nasal polyps was higher in obese subjects than in normal-weight ones [79]. There is a well-known association between CRS and bronchial asthma (BA) [80]. CRS with nasal polyposis is characterized by a type 2 immune response with tissue eosinophilia and high local IgE levels. Lower airway pathologies such as BA are common comorbidities in these patients, and asthma is often more difficult to control [80]. In asthmatic patients, nasal polyposis is associated with increased eosinophilic airway inflammation and worse lung function on spirometry [81]. The shared pathophysiology of CRS and BA has important implications for the therapeutic management of these conditions. It provides a rationale for systemic treatment with novel biologic drugs targeting shared type 2 inflammatory pathways such as Dupilumab and Omalizumab [82,83].

## 5. Obesity and Atopic Dermatitis

Obesity in early childhood is associated with an increased risk of and severity of atopic dermatitis (AD) [84,85,86,87,88]. Kusunoki and colleagues conducted a study to investigate whether childhood obesity is associated with allergic diseases different from asthma. They administered a questionnaire to the parents of 50,086 Japanese schoolchildren. A significant association was found between higher body mass index (BMI) and atopic dermatitis (*p* = 0.03). In addition, children who were obese and had atopic dermatitis were significantly more likely to have severe symptoms [76]. These data are confirmed by the study of Koutroulis which highlights a correlation between the increase in BMI and severity of AD in children older than 2 years [87]. A recent study by Silverberg et al. identified prolonged obesity (>2.5 years) in early childhood as a modifiable risk factor for AD in children [89]. Short-term obesity was not associated with increased AD, suggesting that the rapid correction of obesity with weight loss may be essential for preventing and treating atopic dermatitis in children [89]. Several studies have also been conducted on the adult population, showing a positive correlation between obesity and the risk of developing atopic dermatitis [90,91]. A chronic low-grade inflammatory state induced by adipose hypertrophy in obesity is probably involved in the pathogenesis of AD [91]. The increased expression of proinflammatory leptin and the decreased expression of anti-inflammatory adiponectin in obese patients may represent the pathogenetic link between obesity and AD [86,87,88,89,90,91,92]. Jaworek and colleagues measured the blood levels of different adipokines in a cohort of nonobese adult patients suffering from chronic childhood-onset atopic dermatitis. They found that leptin levels increased while adiponectin levels decreased in patients with AD compared to healthy subjects, and the adiponectin levels were inversely correlated with the disease severity [93]. However, the direct role of adipokines in AD pathogenesis has yet to be fully understood. Bapat and colleagues studied two models of mice with atopic dermatitis. They found that lean and obese mice showed markedly different immune responses. Obesity converted the classical Th2-dependent inflammation associated with atopic dermatitis into a more severe form of the disease with predominant Th17 inflammation. They also noted a different response to biologic therapies targeting Th2 cytokines, which protected lean mice but worsened disease in obese mice [94]. This could be important to understand, to establish effective therapeutic targets for obese children with allergic diseases. Patients with predominant Th1 or Th17 inflammatory responses tend to be less responsive to traditional therapies. Another mechanism that could explain the link between obesity and AD concerns the alterations of the blood-lipid profile typical of obese patients. A recent study found that children with AD had significantly higher levels of total cholesterol (TC) and triglycerides (TG), and this is associated with the Scoring Atopic Dermatitis (SCORAD) index, indicating that dyslipidemia may contribute to AD pathogenesis in children [95]. As regards mechanical factors, skin maceration and stretch marks often observed in obese patients could also favor the development of AD. Excessive subcutaneous adipose tissue could negatively affect the barrier functions of the epidermis. An Italian study investigated the impact of obesity on skin permeability status in children. It was found that skin transepidermal water loss values were significantly higher in obese children compared to normal-weight ones [96]. These results suggest that obesity may contribute to the severity of AD by impairing the skin barrier integrity, therefore allowing allergen entry, and promoting an allergic response that leads to AD. Large prospective cohort studies are still required to confirm the association between AD and obesity.

## 6. Obesity and Food Allergies

Different studies show that obesity might contribute to the increased prevalence of food allergies in children and adolescents. The study by Hayashi and colleagues examined the association between being overweight and the prevalence of food allergies among Japanese children. They analyzed data derived from a self-administered questionnaire from 1772 Japanese children. Children were divided into two groups according to their BMI, overweight and normal weight, and were separated into four age groups to examine age-related differences. They performed univariate and multivariate analyses to examine the association between obesity and FA. In girls, being overweight was significantly associated with FA (OR 1.99, 95% CI 1.01–3.89, *p* = 0.046), while no significant difference was found in boys [97]. Dietary lipids can influence innate immunity functions and antigen presentation to cells of adaptive immunity. Lipids seem to modify the immunostimulating properties of proteins and alter their intestinal absorption, thus modifying the allergen bioavailability [98]. Visness and colleagues found a correlation between C-reactive protein levels and total IgE levels, atopy, and food sensitization [99]. This result suggests that the proinflammatory status observed in obese children could promote the development of food allergies. Some studies have hypothesized that damage to the gastrointestinal barrier induced by a high-fat diet, hyperglycemia, and chronic low-grade systemic inflammation could explain the association between obesity and food allergies [100,101,102]. These variations in gut homeostasis seem to impair tolerance to luminal antigens, thereby increasing susceptibility to food allergies. The pathophysiological mechanisms underlying this phenomenon are not yet fully understood. Torres and colleagues investigated changes in the intestinal mucosa of diet-induced obese mice, finding that they showed increased gut permeability and a reduced number of intraepithelial Tregs. After oral ovalbumin (OVA) administration, obese mice failed to develop oral tolerance and showed a more severe food allergy when exposed to OVA [102]. Changes in the intestinal microbiota also appear to be involved in the development of obesity and food allergies [103,104]. Further prospective studies are necessary to find the causal relationship between obesity and FA.

## 7. Obesity and Chronic Urticaria

In recent years, different studies have investigated the influence of being overweight and obesity on the development of CU. A recent Italian study analyzed the different risk factors associated with CU and showed that the risk of CU is statistically significantly higher in the presence of obesity [105]. This association is probably due to adipokine-mediated inflammation. The chronic, systemic low-grade inflammatory state of the obese can decrease immunological tolerance to antigens, thus increasing the risk of CU [105]. Ye and colleagues analyzed the different components of metabolic syndrome in patients with CU and healthy controls, revealing that central obesity was more prevalent in subjects with CU and significantly correlated with levels of TNF-alpha, serum total IgE, and eosinophil cationic protein [106]. Zbiciak and colleagues enrolled 85 patients with CU, finding an association between CU and heavier weight, higher BMI, and older age at disease onset. Subjects with a higher BMI also tended to have longer disease duration [107]. Other studies suggest that a high waist circumference rather than high BMI is the major risk factor for long-lasting CU [108]. Further analyses to confirm the presented results and possible association between obesity and CU occurrence are needed.

## 8. Conclusions

Obesity is a major risk factor for allergic diseases and is at least partly responsible for their recent burden. Several epidemiological studies have described excess adiposity as a risk factor for asthma. Mechanical and inflammatory effects of obesity can lead to asthma development. Excess adipose tissue in the chest wall and abdomen reduces the functional residual capacity (FRC) and forced vital capacity (FVC) of the lung, leading to increased airway resistance and increased work when breathing, with decreased lung compliance. Impairments of innate and acquired immunity in obese patients also affect asthma development and severity. The polarization toward Th1 and Th17 immune response and the increase in proinflammatory macrophages (M1) lead to increased asthma severity and explain why currently available asthma medications, including corticosteroids, leukotriene inhibitors, and biological agents against the Th2 response and eosinophils, are less effective in obese patients with asthma. Excess adiposity is associated with increased production of inflammatory cytokines (IL-6, IL-1ß, and TNF-alpha), and adipokines (leptin and resistin), leading to low-grade systemic inflammation and increased risk of asthma exacerbations. The exact mechanism for the association between obesity and other atopic diseases remains unclear. Several studies show an association between the chronic low-grade systemic inflammation typical of obese patients and the development of allergic rhinitis, nasal polyposis, atopic dermatitis, food allergies, and chronic urticaria. Vitamin D deficiency appears to play a role in allergic rhinitis, while dyslipidemia and skin barrier defects could explain the link between obesity and atopic dermatitis. Further studies on the association between adiposity and atopy are needed, and they may confirm the biologically active role of fat tissue in the development of allergic diseases. Considering the complex mechanical and immunological mechanisms linking obesity and allergic diseases, therapy alone cannot be sufficient to treat these patients. Starting from this evidence, it becomes of fundamental importance to act on body weight control to achieve general and allergic health, disentangling the detrimental link between obesity allergic diseases and childhood obesity. Obese asthmatic children with dyslipidemia or vitamin D deficiency might benefit from diet, physical exercise, vitamin D supplementation, and cholesterol-lowering drugs. Further studies on the association between adiposity and atopic diseases are needed, confirming the biologically active role of fat tissue in the development of allergic diseases and exploring the possibility of new therapeutic strategies. The new knowledge in inflammation pathways and specific mediators might allow for the development of new treatments for patients with pediatric obesity.

## Data Availability

Not applicable.

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
