# Peer review of "Allergic Diseases and Childhood Obesity: A Detrimental Link?"

_biomedicines, 2023, doi:10.3390/biomedicines11072061_

Round 1

Reviewer 1 Report

1. Figure 1 is too simplistic. Please change the expression to something more informative drawing to the reader.

2. Please, add the paragraph regarding "Obesty and allergic nasal polyp". 

Author Response

  1. Figure 1 is too simplistic. Please change the expression to something more informative drawing to the reader.

We deleted figure 1 because it adds nothing to the text.

  1. Please, add the paragraph regarding "Obesty and allergic nasal polyp."

We have added the paragraph that you suggested.

Reviewer 2 Report

The manuscript summarizes the relationship between obesity and allergic diseases, and results from epidemiological and clinical trials suggest that obesity may aggravate the development of allergic diseases, especially asthma,but this association seems to be more conflicting for allergic rhinitis, atopic dermatitis, and chronic urticaria. This manuscript also briefly illustrates the potential mechanism of obesity as a risk factor for allergic diseases, and provides evidence to disentangle this “detrimental link” between patients with obesity and atopic diseases. The language of this manuscript is well organized, and the summary of related literature is relatively comprehensive. So, it is recommended to accept it after  revision. The detailed comments are as follows:

1. Abstract is an important part of the paper and should reflect the key findings. It is suggested to add important conclusions in the abstract.

2. The logical relationship between lines 35-37 of the manuscript is confused, which makes the reader very confused about the specific content to be expressed. Please modify it.

3. The contents of lines 49-56 are not consistent with the contents of the title. It is suggested to delete and make supplement that allergic diseases are the result of the interaction of genetic and environmental factors. When the gene sequence remains unchanged, environmental factors (obesity) may aggravate the susceptibility to allergic diseases through epigenetic modification.

4. What does the cell in the center of figure 1 represent? Please clearly express what you want to say in the annotations in figure 1.

5. In the fifth part, the related research of Obesity and food allergies is not comprehensive enough, so it is suggested to make a supplement .

6.The format of the references is not uniform. Please check and revise it carefully.

No comments.

Author Response

The manuscript summarizes the relationship between obesity and allergic diseases, and results from epidemiological and clinical trials suggest that obesity may aggravate the development of allergic diseases, especially asthma,but this association seems to be more conflicting for allergic rhinitis, atopic dermatitis, and chronic urticaria. This manuscript also briefly illustrates the potential mechanism of obesity as a risk factor for allergic diseases, and provides evidence to disentangle this “detrimental link” between patients with obesity and atopic diseases. The language of this manuscript is well organized, and the summary of related literature is relatively comprehensive. So, it is recommended to accept it after  revision. The detailed comments are as follows:

  1. Abstract is an important part of the paper and should reflect the key findings. It is suggested to add important conclusions in the abstract.

We modified the conclusion in the abstract.

  1. The logical relationship between lines 35-37 of the manuscript is confused, which makes the reader very confused about the specific content to be expressed. Please modify it.

We agree with the reviewer. We made the changes requested.

  1. The contents of lines 49-56 are not consistent with the contents of the title. It is suggested to delete and make supplement that allergic diseases are the result of the interaction of genetic and environmental factors. When the gene sequence remains unchanged, environmental factors (obesity) may aggravate the susceptibility to allergic diseases through epigenetic modification.

We agree with the reviewer. We made the changes requested.

  1. What does the cell in the center of Figure 1 represent? Please clearly express what you want to say in the annotations in Figure 1.

We deleted Figure 1 because it adds nothing to the text.

  1. In the fifth part, the related research on Obesity and food allergies is not comprehensive enough, so it is suggested to make a supplement.

We made a supplement to this section.

  1. The format of the references is not uniform. Please check and revise it carefully.

We corrected it.

Reviewer 3 Report

This is an extensive review of an important topic, the potential relation between overweight-obesity and allergic diseases. My problem is, if this review has new information. Many reviews are written, a recent review is from Reyes-Angel, lancet child adelesc 2022. What is new in this review?

So, it is a good review, but does it contain new information or a new viewpoint?

Author Response

This is an extensive review of an important topic, the potential relation between overweight-obesity and allergic diseases. My problem is, if this review has new information. Many reviews are written, a recent review is from Reyes-Angel, lancet child adelesc 2022. What is new in this review? So, it is a good review, but does it contain new information or a new viewpoint?

This review contains a new holistic viewpoint. Based on our research in the literature, the theme of the link between obesity and a single atopic disease (i.e., asthma, allergic rhinitis) is deepened. On the other hand, a holistic review involving all childhood atopic diseases is not present in the literature. This review can help the reader to develop a complete and non-fragmentary idea of the link between childhood obesity and all childhood atopic diseases.

Reviewer 4 Report

Allergic diseases and childhood obesity: a detrimental link?

Camilla Stefani1, Luca Pecoraro1, Carl-Erik Flodmark2, Marco Zaffanello1, Giorgio Piacentini1, Angelo Pietrobelli1,3 4

Comments

1-      This is an interesting review however, I’m not sure whether the proposed topic of childhood obesity is really met. Some sections of this review talk about obesity in general and we lose the link with childhood diseases.  I understand this is probably because the data available comes from adult patients however, it makes this review less specific than intended so my suggestion is really focus on the issue of child obesity, redirecting the data as much as possible to this topic.

2-      While it is acceptable to cite other reviews, you should not have too many of them on your reference list. As a review author, you are expected to read the original studies and not rely too much on what other reviews have said or on their interpretation.  I expect to see your own interpretation of the original data and there is no good reason why this has not been done (reference 4, 7, 9, 30, 38, 47, 49, and others).

3-      There are some important experimental studies trying to elucidate the link between obesity and allergic diseases. I feel the authors could have made a better use of the data available to enrich this review. For example, 10.14814/phy2.13082; doi: 10.1289/ehp.9780; doi: 10.3390/nu11122902 and others.

4-      “Epidemiological studies have demonstrated the connection between asthma and obesity. Although the relationship between asthma and obesity remains unclear, many authors suggest that obesity is a risk factor for asthma” This sentence is confusing, I suggest rephrasing. If a connection between asthma and obesity has been demonstrated, then how come that the relationship between both conditions remain unclear?  The authors mean the underlying mechanisms of this connection are not clear?

5-      There are some spelling errors that need correction.

There are some spelling errors in the text, please check carefully

Author Response

  1. This is an interesting review however, I’m not sure whether the proposed topic of childhood obesity is really met. Some sections of this review talk about obesity in general and we lose the link with childhood diseases. I understand this is probably because the data available comes from adult patients however, it makes this review less specific than intended so my suggestion is really focus on the issue of child obesity, redirecting the data as much as possible to this topic.

This review aims to help the reader develop a complete and non-fragmentary idea of the link between childhood obesity and all childhood atopic diseases. We also used data from adult patients to achieve this aim. Anyway, We think that the pediatric focus of the article has never been lost. The conclusions of the article also emphasize the role of the pediatrician.

  1. While it is acceptable to cite other reviews, you should not have too many of them on your reference list. As a review author, you are expected to read the original studies and not rely too much on what other reviews have said or on their interpretation. I expect to see your own interpretation of the original data and there is no good reason why this has not been done (reference 4, 7, 9, 30, 38, 47, 49, and others).

The choice to consider some reviews derives from the need to develop a state of the art on each pediatric allergic disease and its link with childhood obesity. In general, I agree with your suggestion to interpret all original data in the literature for each allergic disease. The authors have chosen to consider other reviews in order not to make the article too dispersive and create confusion in the reader. Anyway, I think the literature summary is relatively comprehensive (86 references), and we added other references based on original articles (see below).

  1. There are some important experimental studies trying to elucidate the link between obesity and allergic diseases. I feel the authors could have made a better use of the data available to enrich this review. For example, 10.14814/phy2.13082; doi: 10.1289/ehp.9780; doi: 10.3390/nu11122902 and others.

We added these references following your suggestion.

  1. “Epidemiological studies have demonstrated the connection between asthma and obesity. Although the relationship between asthma and obesity remains unclear, many authors suggest that obesity is a risk factor for asthma” This sentence is confusing, I suggest rephrasing. If a connection between asthma and obesity has been demonstrated, then how come that the relationship between both conditions remain unclear? The authors mean the underlying mechanisms of this connection are not clear?

We modified this sentence.

  1. There are some spelling errors that need correction.

Comments on the Quality of English Language: There are some spelling errors in the text, please check carefully.

Grammatical errors were addressed using the premium version of the software "Grammarly."

Round 2

Reviewer 2 Report

It is very interesting to elucidate the link between obesity and allergic diseases. In this manuscript, the questions raised previously are well responded. However, some important experimental studies are needed in the manuscript to attempt to elucidate the potential mechanisms underlying the connection between obesity and allergic diseases.

Author Response

We thank the reviewer and agree with his/her comment. We have added it in the section “conclusions”.

Reviewer 4 Report

The authors have introduced original citations and have added some experimental work that improves their review. However,  I am not entirely satisfied with their response to my comments for reasons stated below. 

  1. This is an interesting review however, I’m not sure whether the proposed topic of childhood obesity is really met. Some sections of this review talk about obesity in general and we lose the link with childhood diseases. I understand this is probably because the data available comes from adult patients however, it makes this review less specific than intended so my suggestion is really focus on the issue of child obesity, redirecting the data as much as possible to this topic.

This review aims to help the reader develop a complete and non-fragmentary idea of the link between childhood obesity and all childhood atopic diseases. We also used data from adult patients to achieve this aim. Anyway, We think that the pediatric focus of the article has never been lost. The conclusions of the article also emphasize the role of the pediatrician.

My comment: It is the role of the reviewer to draw the authors' attention to any potential issue that may occur in their manuscript.  if the reviewer is pointing it out to a loss of connection between the data and the subject in some section then there is a very good chance that this is probably the case and the authors should take that observation more seriously.

  1. While it is acceptable to cite other reviews, you should not have too many of them on your reference list. As a review author, you are expected to read the original studies and not rely too much on what other reviews have said or on their interpretation. I expect to see your own interpretation of the original data and there is no good reason why this has not been done (reference 4, 7, 9, 30, 38, 47, 49, and others).

The choice to consider some reviews derives from the need to develop a state of the art on each pediatric allergic disease and its link with childhood obesity. In general, I agree with your suggestion to interpret all original data in the literature for each allergic disease. The authors have chosen to consider other reviews in order not to make the article too dispersive and create confusion in the reader. Anyway, I think the literature summary is relatively comprehensive (86 references), and we added other references based on original articles (see below).

My comment: This  response from the authors makes no sense. If the authors are aiming to present the "state of the art" on each disease, then this is even more of a reason to discuss original work and not reviews. Besides, how citing original work in a review would "confuse" the reader? if the discussion is focused and well made, there is absolutely no reason why it should create confusion. Besides,this is what reviews are for, to bring to the reader the newest data produced in the field and discuss it, not to repeat what it has already been said. It does not matter if the reference list contain 86 citations or 860, it is not the number of citations that makes a review relevant or comprehensive but the quality of these citations and is the author's duty to make sure this is the case. 

Author Response

We are sorry that the reviewer thinks we do not take his words seriously. We have explained our point of view and the reference to the pediatric age is present in the introduction, conclusion, and almost every chapter. We also think removing all non-pediatric references from the article would weaken it. Anyway, we have tried to improve the quality of the bibliographic references, adding further pediatric original articles related to the reviews already mentioned. We remain available for suggestions on specific parts of the article that should be modified to align with the reviewer's idea.

Round 3

Reviewer 2 Report

It is well responded